# Bathing Water Quality Monitoring Practices in Europe and the United States

**DOI:** 10.3390/ijerph18115513

**Published:** 2021-05-21

**Authors:** Ananda Tiwari, David M. Oliver, Aaron Bivins, Samendra P. Sherchan, Tarja Pitkänen

**Affiliations:** 1Expert Microbiology Unit, Finnish Institute for Health and Welfare, P.O. Box 95, FI-70701 Kuopio, Finland; tarja.pitkanen@thl.fi; 2Biological and Environmental Sciences, University of Stirling, Stirling FK9 4LA, UK; david.oliver@stir.ac.uk; 3Department of Civil & Environmental Engineering & Earth Science, University of Notre Dame, 156 Fitzpatrick Hall, Notre Dame, IN 46556, USA; abivins@nd.edu; 4Department of Environmental Health Sciences, Tulane University, 1440 Canal Street, New Orleans, LA 70112, USA; sshercha@tulane.edu; 5Department of Food Hygiene and Environmental Health, Faculty of Veterinary Medicine, University of Helsinki, FI-00014 Helsinki, Finland

**Keywords:** bathing and recreational water, microbial quality, tool-box-approach, bathing water directive, recreational water quality criteria

## Abstract

Many countries including EU Member States (EUMS) and the United States (U.S.) regularly monitor the microbial quality of bathing water to protect public health. This study comprehensively evaluates the EU bathing water directive (BWD) and the U.S. recreational water quality criteria (RWQC) as regulatory frameworks for monitoring microbial quality of bathing water. The major differences between these two regulatory frameworks are the provision of bathing water profiles, classification of bathing sites based on the pollution level, variations in the sampling frequency, accepted probable illness risk, epidemiological studies conducted during the development of guideline values, and monitoring methods. There are also similarities between the two approaches given that both enumerate viable fecal indicator bacteria (FIB) as an index of the potential risk to human health in bathing water and accept such risk up to a certain level. However, enumeration of FIB using methods outlined within these current regulatory frameworks does not consider the source of contamination nor variation in inactivation rates of enteric microbes in different ecological contexts, which is dependent on factors such as temperature, solar radiation, and salinity in various climatic regions within their geographical areas. A comprehensive “tool-box approach”, i.e., coupling of FIB and viral pathogen indicators with microbial source tracking for regulatory purposes, offers potential for delivering improved understanding to better protect the health of bathers.

## 1. Introduction

Visiting bathing sites located around lakes, rivers, estuaries, and coastal areas are major summertime recreational activities and provide a range of physical and psychological health benefits [1]. Further, such activities make a large contribution to revenue collection and employment generation through coastal tourism. For example, such tourism represents ~80% of all tourism and ~50% of international tourism [2]. However, the microbial contamination of bathing sites from various sources such as sewage effluents, agricultural runoff, and accidental releases from municipal sewage sources poses a serious public health risk and jeopardizes the health and economic benefits associated with bathing [3,4].

Many countries such as European Union Member States (EUMS), other European countries e.g., the United Kingdom (U.K.), the United States (U.S.), Australia, and Canada are regulating the microbial contamination of recreational and bathing waters. These counties regularly monitor bathing sites to protect public health by ensuring safe water quality [5,6]. This study compares the bathing water quality management and monitoring practices associated with the EU bathing water directive (BWD) and the U.S. recreational water quality criteria (RWQC). This study considers the strengths and limitations of currently used European and U.S. regulatory frameworks and raises awareness of best practices. Further, this study presents future perspectives of bathing water quality management based on existing scientific evidence and knowledge.

### 1.1. Public Health Risks and Etiological Agents

Recreational exposure to contaminated water increases the risk of waterborne illnesses such as diarrhea, respiratory illness, skin rashes, fever, ear, and eye infection [7,8,9,10,11,12,13,14,15]. The possible types of illness are primarily determined by the etiological agents and their mode of exposure (Table 1). The majority of etiological agents for bathing-related illnesses, such as *Campylobacter* spp., *Salmonella* spp., *E. coli* O157: H7 and *E. coli* O111, *Shigella* spp., adenovirus, norovirus, poliovirus, coxsackievirus, echovirus, *Giardia lamblia,* and *Cryptosporidium parvum* originate from human and animal fecal sources [7,11,16,17,18,19,20]. Among them, human infecting viruses are mostly host-specific and originate from human fecal contamination [16,21], and some other bacterial pathogens and protozoan parasites such as *C. jejuni*, *E. coli* O157: H7, *Salmonella* spp., *Giardia lamblia,* and *Cryptosporidium parvum* originate from both animal and human fecal sources [3,7,22]. Naturally occurring aquatic microbes, such as toxin-producing *Cyanobacteria* spp. and pathogenic *Vibrio* spp. are also a major cause of bathing-related illnesses [20,23,24].

### 1.2. Faecal Indicator Bacteria 

The direct enumeration of enteric pathogens within bathing waters provides the most specific evidence of their presence at bathing sites. However, doing so consistently for regulatory purposes is not practically or economically feasible with currently available methodologies due to the diversity of enteric pathogens in bathing water and their associated enumeration protocols [21,25]. Therefore, fecal indicator bacteria (FIB), a set of commensal gut microbes from warm-blooded animals, have been enumerated for decades, as indicators of feces and subsequently enteric pathogens. FIB are present in high numbers in the gut of warm-blooded mammals and pass through the same transmission route as enteric pathogens [18]. Enumeration of FIB is currently the most widely accepted approach for monitoring the microbial quality of bathing water [5,6,7,15,25,26,27,28].

The positive relationship between bathing illness episodes and viable counts of FIB, i.e., *E. coli*, mainly in freshwater, and enterococci (ENT), in both fresh and marine water, has been reported [10,29,30]. However, such a positive relationship was not found in bathing sites with non-point sources of contamination [9,31]. The variable relationship between FIB counts and bathing illness episodes may reduce the reliability of FIB and may complicate the interpretation of FIB counts for assessing human health risk [9]. 

There can be many reasons for a weak relationship between FIB and bathing-related illness episodes. FIB are common in the gut of all warm-blooded animals, whereas pathogens are common only in infected hosts. Further, even if the FIB and enteric pathogens originate from the same host, their decay rates are different in the environment [32,33,34,35,36]. The wide taxonomic ranges of enteric microbes have different cell structures, morphology, and physiology. Thus, each of them responds differently to environmental stress factors such as pH, solar radiation, salinity, predation, temperature, and nutrients [11,24,32,37]. Therefore, successful FIB detection and enumeration (*E. coli* and intestinal enterococci) does not necessarily imply the presence of pathogens from wide taxonomic ranges of bacteria, protozoa, and viruses [18,25,38]. For example, a laboratory study [of seeded coastal water] demonstrated a strong positive correlation between MS2 coliphage and *Enterococcus faecalis* only during the first few days of the experiment [36]. Other studies detected enteric viruses in surface water even when the FIB numbers were below the safe limit according to current monitoring protocols [16,39]. Other epidemiological studies have demonstrated stronger relationships between gastrointestinal (GI) illness and F^+^ RNA coliphages and somatic coliphages than with FIB [30,38]. 

Growth or persistence of enteric microbes and FIB in environmental habitats like soil, sediments, vegetation, and fecal matrices have been reported [37,40,41,42,43,44]. The contamination of bathing water from such environmental sources could also weaken the relationships between FIB and enteric pathogens and bathing water-related illness episodes. Further, the illnesses due to autochthonous microbes like *Vibrio* spp. and *Cyanobacteria* toxins have almost no relationship with FIB counts [20,23,24].

### 1.3. Monitoring of FIB

For regulatory purposes, culture-based methods are almost unanimously used as the method for enumerating FIB in bathing water [5,6]. Culture-based methods are affordable, highly standardized, and easy to operate [45,46,47]. These methods enumerate the viable and culturable cells of FIB; thus, these may relate well with viable pathogens that are a concern of human health risk [47,48,49]. However, these methods are criticized as being unable to enumerate viable but non-culturable cells and injured cells, which may recover their viability under favorable conditions [50]. Culture-based methods are time-consuming and require additional time (~24–48 h) to produce results due to the need for incubation and growth of the target microbes [51,52]. Further, the identification criteria of these methods are based on phenotypic characteristics such as colony morphology and color. Sometimes, identifications based on the phenotypic character are subject to user bias [45,48,49].

As a limitation, the FIB enumerated with culture-based methods cannot differentiate between sources of contamination, and subsequently, an equal level of human health risk is inferred from all contamination source types [21,53,54]. Although, the fecal source constrains the possible presence of etiological agents (Table 1), and subsequently the level of human health risk. For example, human-infecting viruses are mostly found in human fecal contamination [3,9,21,55], and zoonotic pathogens like *Salmonella* spp., *Campylobacter* spp., pathogenic *E. coli*, *Cryptosporidium*, *Giardia*, *Leptospira*, and *Brucella* spp. are common in infected hosts, in both animal and human fecal materials [3,22,56]. 

Culture-independent methods such as real-time polymerase chain reaction (qPCR) and quantitative reverse transcription-polymerase chain reaction (RT-qPCR) overcome some of the limitations of culture-based methods [57]. The qPCR methods have higher specificity than the culture-based methods [33]. Sources of contamination can be differentiated by using qPCR primers developed from host-specific bacteria such as *Bacteroides*, *Catellicoccus*, and *Brevibacterium*, in the microbial source tracking (MST) process [21,53,54,58,59,60,61]. The combination of MST with the current FIB monitoring process can make it easier and simpler for interpreting colony counts and microbial risk assessment by predicting potential pathogens [21,54]. 

The qPCR-based methods enumerate target microbes more rapidly than culture-based methods [62,63,64]. The U.S. Environmental Protection Agency (USEPA) has allowed the use of qPCR methods as a rapid method since 2012 for monitoring bathing water for regulatory purposes [6]. However, these methods also have important limitations. For example, the most frequently used taxonomic genes, 16S rRNA, may not be able to distinguish closely related species in some genera where the genes are highly conserved [46,65]. Further, gene copies enumerated with this method can be affected by the PCR amplification efficiency, the detection limit of the assay, volume of the water sample, and the efficiency of the pre-processing steps such as filtration and nucleic acid extraction [50,62,63,65]. The qPCR method can also greatly overestimate the viable microbial counts compared to the culture-based methods as the molecular method may enumerate the total DNA copies from viable, viable but non-culturable, dead cells, and even extracellular nucleic acids [50,62]. 

### 1.4. Bathing Water Guidelines and Regulations 

The bathing water management guideline of the World Health Organization (WHO), the BWD and the RWQC are three major global guidelines and regulations for monitoring bathing water quality [5,6,7,15]. All three guidelines and regulations aim to protect the health of bathers by assessing the microbial quality of bathing sites. Besides human health, the latter two also aim to preserve, protect, and improve the quality of the aquatic environment. All three utilize culture-based FIB enumerations (*E. coli* or enterococci, or both) with counts of FIB above a certain benchmark value signaling increased risk to human health. Such benchmark values are set without regard for the sources of microbial contamination, including sites located in all climatic regions within their political jurisdiction, and in some cases use the same values for determining inland and coastal water quality standards [7]. The approach of using the same indicator for all contamination source scenarios, geographical regions, and bathing site types was defined as a “One-Size-Fits-All” approach by USEPA (2007) [66]. However, the BWD has set different benchmark values for FIB enumerated at inland and coastal bathing sites [5]. 

The WHO guideline asserts that public health risk due to contaminated bathing sites can be best assessed by combining a qualitative sanitary survey and quantitative FIB data. It recommends grading bathing sites on five categories (very good, good, fair, poor, and very poor) based on the sanitary survey and FIB counts (Figure 1). The sanitary survey collects information about sewerage and stormwater pipe networks, possible microbial contamination sources, information on historical contamination, and comparisons to detected FIB counts. It assigns the highest health risk levels for contamination from human fecal sources. It recommends analyzing at least 20 water samples during each bathing season of the year for FIB counts and calculating the percentile value of FIB data for reporting and making design about bathing water quality (Table 2). The WHO guidelines [7,15] recommends intestinal enterococci (iENT) for both inland and coastal water with the same guideline value of 500 colony-forming unit (CFU) or most probable number (MPN)/100 mL (Table 2), with a maximum of 1/10 beachgoers experiencing bathing-associated waterborne infection at least once per year, as estimated in earlier studies [67,68,69]. However, the understating regarding microbes in bathing water has been improved continuously over the last twenty years; and guidelines are updated based on the new information [28,70]. 

The BWD is a legal instrument for monitoring bathing water quality within EUMS [5] and is adopted by some other European countries as well. Monitoring bathing sites generates information about the general status of bathing water and allows reporting to the EU regulatory body [5]. The EU approach is underpinned by the WHO perspective, as both approaches were based on the same epidemiological studies [67,68,69]. As recommended by the WHO approach, the EU approach provides classifications (excellent, good, sufficient, and poor) of bathing sites based on FIB counts determined from the last four-year seasons of monitoring. The excellent and good class is classified based upon the 95th percentile value and the sufficient and poor class is classified based upon the 90th percentile value (Table 2). Further, the EU approach has a provision for making profiles of each bathing site separately, based upon the possible source of contamination, locations, and land-use pattern in the watershed [5]. The BWD allows *E. coli* as a FIB for monitoring inland sites [5]. As of April 2021, the European Commission has initiated an evaluation of the BWD, with completion of this review due by 2023 [70]. 

The RWQC (2012) is the guiding document for monitoring the microbial quality of designated recreational water in the U.S. located in marine, estuarine, the Great Lakes, and inland areas [6]. It aims to protect public health during primary contact (swimming, bathing, surfing, water skiing, tubing, and skin diving) with recreational water by complementing the Clean Water Act. The RWQC (2012) prescribes counts of *E. coli* and ENT in freshwater and ENT in marine water as FIB (Table 2), with culture-based methods [6]. Early studies reported ENT are a good predictor of GI illness in fresh and marine bathing sites and *E. coli* are a good predictor of GI illness in freshwater bathing sites [29,71]. Additional studies [10,14], further verified the use of *E. coli* and ENT for monitoring microbial quality of bathing water. 

### 1.5. Differences between the Current European and U.S. Practices

Although there are parallels in terms of laboratory methods and deployment of regulatory sampling and reporting between the BWD and RWQC, there are some clear differences in these two regulatory frameworks. The major dissimilarities are explained here:


I.Enumeration Methods and Indicators


The principal difference between BWD and the RWQC is the selection of reference methods for the enumeration of FIB. The BWD relies on the international standards (ISO) adopted by the European countries for standardization of approach and the U.S. regulations refer to the methods published by the USAEPA. Further, The RWQC uses the term Enterococci (ENT) and the BWD uses the term intestinal enterococci (iENT), and these two terms are considered equivalent and have been used interchangeably [37], but there are some differences to note.


a.Enterococci or Intestinal Enterococci


The BWD prescribes two reference methods ISO 7899-1 (MPN-based) and ISO 7899-2 (membrane-filtration-based) for selective isolation and enumeration of iENT [5]. The ISO 7899-1 method uses the miniaturized 96-well system premised on iENT capacity to hydrolyze 4-methylumbelliferyl-b-D-glucoside in the presence of thallium acetate, nalidixic acid, and 2,3,5-triphenyltetrazolium chloride, in the liquid medium [51]. The presence of iENT is visualized by the emission of fluorescence in 36–72 h. The second method (ISO 7899-2) is based on membrane filtration and confirms iENT in two steps [52]. First, the bacteria retained on the membrane filter are incubated on Slanetz and Bartley (S&B) medium. The triphenyltrazolium chloride (TTC) in S&B medium is reduced to formazan and forms red colonies. In the second step, the membrane filter is transferred to bile esculin azide (BEA) agar and presumptive colonies are confirmed as iENT. The iENT is confirmed based on dark brown to black colonies produced on BEA agar medium. The taxonomic distribution of iENT with EU methods (ISO 7899-1, ISO 7899-2) includes primarily four species: *Enterococcus faecalis*, *Enterococcus faecium*, *Enterococcus durans*, and *Enterococcus hirae.* Tiwari et al. (2018) observed that 90% of iENT isolated with ISO 7899-2 method were *E. faecalis* and *E. faecium* [46]. 

The RWQC (2012) recommends EPA Method 1600 (mEI agar method) for enumeration of ENT [6]. This method is based on a membrane filtration approach, with a single confirmation step using *Enterococcus* indoxyl-*β*-D-glucoside (mEI) agar as the culture media. ENT are confirmed as blue halo colonies on the media. This method enumerates a broader taxonomic group than EU reference methods, including *E. faecalis*, *E. faecium, Enterococcus casseliflavus*, and *Enterococcus mundtii* [72,73]. Two species, *E. faecium* and *E. faecalis* are considered the most prevalent *Enterococcus* species in human feces [74,75]. *Enterococcus* is a large genus-group and some species, such as *E. mundtii, E. casseliflavus, Enterococcus aquimarinus,* and *Enterococcus sulfureus,* are often associated with vegetation [37,74]. 

Further, the RWQC allows local authorities to use alternative methods that give equivalent FIB counts when compared with the reference method [6]. An earlier study demonstrated that ENT enumerated with EPA Method 1600 method was equivalent with the Enterolert method as an alternative for ENT enumeration at U.S. bathing sites [73]. In addition to culture-based FIB counts, RWQC (2012) also allows the use of gene copies of the 23S rRNA gene of *Enterococcus* spp. as a rapid enumeration method [6,62].


b.
*Escherichia coli*



The BWD prescribes two reference methods ISO 9308-1 2000 (membrane filtration based) and ISO 9308-3 1998 (based on miniaturized most probable number) for selective isolation and enumeration of *E. coli* [5,76,77]. Previously, the ISO 9308-1 method operated on two steps using TTC Tergitol^®^ 7 agar and rapid test using TSA/TBA agar [76]. The ISO 9308-1 method has been modified to ISO 9308-1 2014 based on the chromogenic definition of *E. coli* with Chromogenic Coliform Agar (CCA) media [78]. Both versions ISO 9308-1 (2000, 2014) have been criticized as being unsuitable for monitoring environmental waters containing high levels of background bacterial flora [28,45,79]. The ISO 9308-3 method detects *E. coli* based on a fluorogenic reaction (positive for β-glucuronidase) [47,80]. ISO 9308-3 is a robust and reliable method for use with surface and wastewater samples. It uses a 10 mL sample volume which can be too small for bathing water samples when *E. coli* counts are lower [77]. This method can have a higher false-positive rate than ISO 9308-1 [45].

EUMS are permitted to use an alternate method for enumeration of FIB that gives equivalent counts with reference methods based on the ISO 17994 criteria [45,47,81]. The ISO 9308-2 2012 (Colilert-18 Quanti-Tray) method has produced equivalent *E. coli* counts with the reference methods and this method can be used for enumeration of *E. coli* for regulatory purposes in many EUMS [45,47,80]. This method detects *E. coli* based on a fluorogenic reaction (positive for β-glucuronidase) [82]. This method returns results more rapidly (18 h) than the reference methods (48–72 h). 

The RWQC (2012) recommends using EPA Method 1603 for enumerating *E*. *coli* during bathing water quality monitoring in the U.S. [6]. This method is based upon a membrane filtration approach, with a single confirmation step using modified mTEC agar [72,73]. *E. coli* is confirmed as red or magenta color colonies. Similar to the EU method, the defined substrate method (Colilert-18) produces equivalent *E. coli* counts with U.S. reference methods. Therefore Colilert-18 also can be used as an alternative method for *E. coli* enumeration in water sampled at U.S. bathing sites [73]. 


II.Epidemiological Studies


When deriving the guideline value of FIB, the EU regulation was based on a randomized control study [8,67,68,69]; whereas, the U.S. standard was based on a prospective cohort study [10,13,14]. In the randomized control study, bathing volunteers were randomly selected and preassigned as either swimmers or non-swimmers. The selection of healthy adult volunteers during such an epidemiological study systematically ignored the heterogeneity of bathers likely present at real bathing sites (e.g., children, old and immunocompromised people). Ethical constraints also need considering with respect to levels of contamination those swimmers are exposed to, e.g., exposure to bathing waters meeting mandatory standards rather than falling below this threshold. During the prospective cohort studies, bathing volunteers were randomly selected on beaches and their bathing activities were followed. 

Epidemiological studies may differ with respect to microbial monitoring methods, settings, definitions of illness, follow-up duration, and sources of contamination at studied bathing sites [8,10,13,14,67,68,69]. Such heterogeneity between and within studies greatly complicates the comparability of findings. 


III.Acceptable risk


The BWD proposes different indicator values for inland or coastal bathing water sites based on earlier reports [67,68,69]. In comparison to indicator bacteria, enteric viruses are suspected to be more resistant to salinity [7,69,71]. Therefore, even with comparable FIB levels, coastal water is likely to have double the likelihood of enteric viruses being present and thus risk of illness [69,71]. For example, the BWD equally accepts bathing water-related infection risk of GI illness and acute febrile respiratory illness (AFRI) for both freshwater and marine water, so the indicator values were varied according to water types. The European criterion accepts 2.5% AFRI and 5% GI illness anually (Table 2). The BWD considers the risk of both acute febrile respiratory illness (acute febrile respiratory illness; fever accompanied by headache body ache usually fatigue or anorexia, sore throat, runny nose or cough) and GI illness (vomiting, diarrhea, nausea, or indigestion). However, the RWQC uses comparable indicator values for both inland and coastal bathing sites and accepts higher risk in coastal bathing sites than inland sites [6]. The maximum acceptable risk of RWQC (2012) for GI illness (vomiting or diarrhea, nausea with a stomachache, nausea, or stomachache) in freshwater is 3.2% and marine is 3.6% per year. The RWQC does not account for respiratory illness. 


IV.Reporting metrics


The BWD and the RWQC use different reporting metrics for the estimation of FIB counts [5,6]. The BWD uses the 95th and 90th percentile of samples collected across four consecutive bathing seasons (i.e., a four-year dataset). The BWD recommends collecting at least four water samples during each bathing season and calculating percentile values of the data from the last four years for the classification of a bathing site. There is much variation in the reporting of samples per bathing season, ranging from 4 to about 20 samples, per bathing season. Greater numbers of samples are analyzed from the bathing sites located nearby larger cities, where more visitors go for bathing each day during the bathing season. The BWD can have a risk of misclassification of the bathing sites when the minimum number of sample numbers, i.e., ~16 samples of the last four years are included for analysis [28]. The maximum acceptable 90th percentile count of *E. coli* is 500 in coastal waters and 900 CFU or MPN/100 in inland bathing waters; for iENT those counts are 185 and 330 MPN/100 mL in coastal and inland bathing waters, respectively [5].

The RWQC requires geometric mean (GM) or statistical threshold value (STV), which is nearly equivalent with 90th percentile, to be met more frequently than the BWD requirement (every 30 days) [6]. In freshwater, the maximum acceptable *E. coli* counts are 100 CFU or MPN/100 mL GM, or 320 CFU or MPN/100 mL STV [6]. While the maximum acceptable ENT counts are 30 CFU or MPN/100 mL GM, or 110 CFU or MPN/100 mL STV, in freshwater, and 35 CFU or MPN/100 mL GM, or 130 CFU or MPN/100 mL STV, in coastal water [6]. The STV value should not be exceeded by more than 10% of the samples used to calculate the GM for informing the public about the microbial quality of bathing water [6].

The GM approach stabilizes the data because it is not as sensitive to extreme observations. In contrast, the 95th or 90th percentile approaches account for the extreme observations and give increased weighting to the variance of the observed data. Sometimes, high 95th or 90th percentile values may result despite lower FIB central tendency due to large variance. Nonetheless, a large variance in FIB counts indicates an increased likelihood of high FIB counts than data with a lower variance. 


V.Profiling and classification


The BWD has a provision of profiling each bathing site based on the source of pollution and information from a sanitary survey. Further, the BWD classifies bathing sites based on the pollution level assessed with FIB counts; excellent, good, sufficient, or poor [5]. The requirement to classify bathing sites means that beach users can access water quality information to help decide which beach to visit. This in turn may incentivize local authorities to do more regarding their management of bathing water environments [41,63,83,84]. The U.S., in contrast, does not have a provision for classifying and profiling bathing sites.

## 2. Limitations of Current Approach and Future Directions

A major limitation with the currently used regulatory frameworks (both the BWD and RWQC) is that enumerating FIB cannot determine fecal contamination sources. The approaches also weigh equally the FIB counts from all possible sources and assign equal risk to human health from all sources. At bathing sites, there can be a wide variety of pathogens from different sources responsible for a single type of illness. Viruses, bacteria, and protozoan parasites are capable of causing GI illness. However, the source of contamination for these three groups of pathogens can be different. For example, human pathogenic viruses causing GI illnesses mostly originate from human fecal sources, while bacteria and protozoa can originate from zoonotic sources in addition to human sources. Neglecting the source of fecal contamination by only using FIB counts may be insufficient to protect human health. 

The current approach also does not consider the different inactivation rates that are likely for different enteric microbes (belonging to virus, bacteria, and protozoa) in different climatic regions and also in inland versus coastal waters. Environmental factors are one of the major determinants of the fate and decay of FIB and enteric pathogens [34,35,36]. Such environmental factors can vary regionally and globally in different climatic zones.

Additionally, the current regulatory practices do not consider the microbiological quality of beach substrates like coastal sand and vegetation. The microbial quality of such substrates can have important public health implications [36,40,85,86], as can beach debris such as marine plastics [87]. Consideration of the microbial quality of coastal sand and vegetation (and associated management) in addition to measuring water quality can further protect public health [44,86].

Lastly, the FIB currently used may not be specific to the presence of viruses and protozoa in bathing water, as currently there is no provision for enumerating viral and protozoan indicators. However, these two groups of microbes are also major causes of bathing-related illnesses [20,88]. Quantitative microbial risk assessment can be a cost-effective approach for developing regional or site-specific recreational water quality criteria [89]. The use of the high-throughput sequencing of taxonomic and functional genes provides insight into the taxonomic and functional profile of the microbial community, potential pathogens, and microbial diversity [57,90]. Further, real-time prediction of FIB concentrations through process-based modelling or the use of statistical modelling to inform management at bathing sites offers potential for complementing future regulatory approaches and can capitalize on nationally available datasets such as rainfall, temperature, catchment characteristics, river discharge, and wave characteristics. [91,92,93]. However, significant within-day variation in FIB concentrations, especially at coastal sites, means that intensive sampling of bathing waters for FIB is a prerequisite for building and testing models [94]. Numerous other potential approaches have been proposed for regulatory monitoring purposes such as measuring chromophoric dissolved organic matter with a spectrophotometer [95,96], and considering the distribution of living and dormant FIB abundance and their residence time in coastal water [97]. However, adopting new monitoring procedures for bathing water quality regulation is challenging. For example, new procedures affect the required number of samples, incur laboratory installation and staff training costs for a new method, and will likely result in an increase in cost per sample [3,66,89]. Thus, applying and combining up-to-date and emerging scientific knowledge on bathing water quality monitoring may protect public health better than with the currently used approach. Further, setting a common global standard operating protocol, i.e., uniform definition of GI illness, microbial enumeration methodologies, and follow-up time, may make it easier for sharing and comparing research findings at the international level.

## 3. Conclusions

No single microbial indicator can predict infectious disease risk consistently in all environments at all times. The best fecal indicator for one bathing site might be not equally good for the next bathing site [38]. Therefore, a comprehensive way of monitoring bathing water quality is likely to offer a better approach. The USEPA (2007) describes a “Tool-Box-Approach” for bathing water quality monitoring [66]. This approach keeps all possible materials and methods as tools (i.e., FIB, MST tools, viral indicators, protozoan indicators, and pathogen indicators) in a virtual box and uses the most appropriate material and methods for a specific bathing site at a particular time. Using this approach, monitoring of bathing water quality can begin by characterizing the FIB counts with the current approach and source of contamination with MST markers [21,60]. After that, based on the source of contamination and FIB status, an additional more specific indicator can be selected from the toolbox. For example, viral pathogens and indicators can be scrutinized when human fecal contamination is suspected via MST, as human pathogenic viruses are solely found in human fecal material. Similarly, protozoan indicators can be scrutinized if zoonotic bacteria or protozoan parasites are suspected due to cattle markers. Coliphage and enterococci can be predictive of GI illness if the dominant source is human [98]. However, spores of *C. perfringens,* male-specific (F+) coliphages, or molecular markers of *Bacteroides* bacteria can be more reliable than FIB if fecal contamination is suspected to have originated from an environmental source [33]. 

Profiling the microbial quality of bathing water of each bathing site based on historic pollution levels and sources of contamination may help inform such a toolbox for that bathing site. While the BWD requires a profile of each bathing site [5], the use of a comprehensive tool-box approach, i.e., coupling of FIB and viral pathogen indicators with MST for targeting different fecal sources, may further strengthen such profiling of bathing waters and deliver better understanding to protect the health of bathers.

## Figures and Tables

**Figure 1 ijerph-18-05513-f001:**
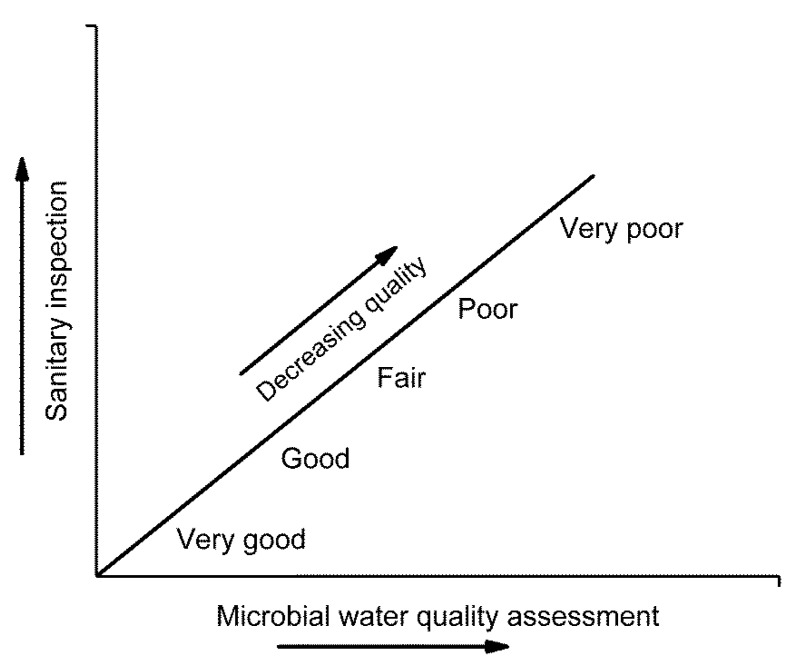
Bathing water health risk classification matrix based on FIB enumeration and sanitary inspection [7].

**Table 1 ijerph-18-05513-t001:** Etiological agents of bathing illness, probable sources and transmission pathways. GI = gastrointestinal [7,19].

Agent	Illness	Probable Source	Transmission Pathway
*Campylobacter* spp.	Gastroenteritis, fever	Human and animals	Ingestion
Enteropathogenic *E. coli*	Bloody diarrhea, abdominal cramp	Human and animals	Ingestion
*Helicobacter pylori*	Gastritis, abdominal pain	Human and animals	Ingestion
*Legionella* spp.	Pneumonia, gastroenteritis	Natural	Inhalation
*Leptospira* spp.	Fever, headache, vomiting, jaundice	Natural and animals	Ingestion
*Salmonella* spp.	Gastroenteritis, fever, pain	Human and animals	Ingestion
*Mycobacterium avium*	Respiratory disease	Natural	Inhalation/contact
*Vibrio vulnificus*	Infection in pre-existed open wound	Natural	Wound infection
*Shigella* spp.	Bacillary dysentery, abdominal pain	Human	Ingestion
Adenovirus	Gastroenteritis, respiratory disease	Human	Ingestion, inhalation
Noroviruses	Gastroenteritis	Human	Ingestion
Rotaviruses	Gastroenteritis	Human	Ingestion
Coxsackievirus	Mild febrile illness to myocarditis	Human	Ingestion
Enteroviruses	Central nervous system, ocular and respiratory infections	Human	Ingestion
Echovirus	Diarrhea, secretions from the eyes or throat	Human	Ingestion
Hepatitis A virus	Liver disease	Human	Ingestion
Hepatitis E virus	Liver disease	Human and animals	Ingestion
*Cryptosporidium*	Diarrhea, abdominal pain, fever	Human and animals	Ingestion
*Giardia*	Diarrhea, abdominal cramp	Human and animals	Ingestion
*Microsporidia*	GI illness, diarrhea	Human and animals	Ingestion
*Naegeria fowleri*	Meningoencephalitis	Natural	Contact
*Schistosoma* spp.	GI illness, haematuria	Human	Ingestion, Contact
*Entamoeba histolytica*	Amoebic dysentery	Human	Ingestion

**Table 2 ijerph-18-05513-t002:** Bathing and recreational water standards, regulations, guidelines, and indicators on freshwater and marine bathing sites. INT Ent = intestinal enterococci, ENT = enterococci, GC = gene copies, STV = statistical threshold value, CCE = calibrator cell equivalents, per = percentile, GM = geometric mean, AFRI = acute febrile respiratory illness, and GI = gastroenteritis [5,6,7].

Regulation or Guideline	Indicator	Water Type	FIB Value (CFU or MPN/100 mL)	Reporting Metric	Illness Rate for Swimmers	Symptoms	Sampling Frequency
[7]	Ent	Fresh/Marine	500	95 per	10% GI illness risk	AFRI, GI illness	~20/per bathing season
[5]	INT Ent	Fresh	200 * (Excellent), 400 * (Good), 330 ** (Sufficient)	* 95 per, ** 90 per	AFRI: Excellent 1%, Good 2.5%, GI: Excellent 3%, Good 5%	AFRI, GI illness	>4/per bathing season
[5]	INT Ent	Marine	100 * (Excellent), 200 * (Good), 185 ** (Sufficient)	* 95 per, ** 90 per	AFRI: Excellent 1%, Good 2.5%, GI: Excellent 3%, Good 5%	AFRI, GI illness	>4/per bathing season
[5]	*E. coli*	Fresh	500 * (Excellent), 1000 * (Good), 900 ** (Sufficient)	* 95 per, ** 90 per	AFRI: Excellent 1%, Good 2.5%, GI: Excellent 3%, Good 5%	AFRI, GI illness	>4/per bathing season
[5]	*E. coli*	Marine	250 * (Excellent), 500 * (Good), 500 ** (Sufficient)	* 95 per, ** 90 per	AFRI: Excellent 1%, Good 2.5%, GI: Excellent 3%, Good 5%	AFRI, GI illness	>4/per bathing season
[6]	ENT	Fresh	30/110 STV	GM/STV	32/1000	GI illness	~5/30 days in bathing season
[6]	*E. coli*	Fresh	100/320	GM/STV	32/1000	GI illness	~5/30 days in bathing season
[6]	ENT	Marine	35/130 STV	GM/STV	36/1000	GI illness	~5/30 days in bathing season
[6]	ENT qPCR (GC)	Fresh/Marine	470 CCE/2000 CCE	GM/STV		GI illness	~5/30 days in bathing season
[6]	ENT qPCR (GC)	Fresh/Marine	1000 CCE	75 per		GI illness	~5/30 days in bathing season

## Data Availability

The study did not report any data.

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
