# Peer review of "Bathing Water Quality Monitoring Practices in Europe and the United States"

_ijerph, 2021, doi:10.3390/ijerph18115513_

Round 1

Reviewer 1 Report

The manuscript deals with a very current and extremely important topic: the status of bathing water quality monitoring to protect public health. The authors report a well structured state-of-the-art of management practices in EUMS and ES, giving a detailed description of the microbiological risk associated with contaminated bathing waters as well as of the regulatory frameworks. However the authors did not consider last advances proposed for bathing water quality monitoring through the development of integrated coastal observing systems which enable in situ real time monitoring of bathing areas and the integration of continous data with remote sensing images and mathematical models to forecast the dispersion and fate of potential pathogens.

Just few suggestions for author's revision:

  1. To improve the readiness and the soundness of the work, I suggest the authors to reduce the manustript of about 20%, trying to avoid repetitions;
  2. The work does not adeguately consider background studies on the possibility to monitor in real time potential contamination sources through  CDOM/tryptophan in situ fluorescence measurements to quickly detect fecal contamination events, thus enabling decision makers to promptly set intervention measures for human health protection. See the following references:Caruso, G.; Denaro, R.; Genovese, M.; Giuliano, L.; Mancuso, M.; Yakimov, M. New methodological strategies for detecting bacterial indicators. Chem. Ecol. 2004, 20, 167–181, doi:10.1080/02757540410001690333. Tedetti, M.; Longhitano, R.; Garcia, N.; Guigue, C.; Ferretto, N.; Goutx, M. Fluorescence properties of dissolved organic matter in coastal Mediterranean waters influenced by a municipal sewage effluent (Bay of Marseilles, France). Environ. Chem. 2012, 9, 438–449, doi:10.1071/EN12081.Moore, C.; Barnard, A.; Fietzek, P.; Lewis, M.R.; Sosik, H.M.; White, S.; Zielinski, O. Optical tools for ocean monitoring and research. Ocean Sci. 2009, 5, 661–684, doi:10.5194/os-5-661-2009.
    Zielinski, O.; Busch, J.A.; Cembella, A.D.; Daly, K.L.; Engelbrektsson, J.; Hannides, A.K.; Schmidt, H. Detecting marine hazardous substances and organisms: Sensors for pollutants, toxins, and pathogens. Ocean Sci. 2009, 5, 329–349, doi:10.5194/os-5-329-2009.
    Madonia, A., Caruso, G., Piazzolla, D., Bonamano, S., Piermattei, V., Zappalà, G., & Marcelli, M. (2020). Chromophoric Dissolved Organic Matter as a Tracer of Fecal Contamination for Bathing Water Quality Monitoring in the Northern Tyrrhenian Sea (Latium, Italy). Journal of Marine Science and Engineering8(6), 430.

  3. A big effort has been made in the last decades in developing mathematical models to properly simulate the fate of pathogens in seawater. (see  Bonamano, Simone, et al. "Development of a New Predictive index (Bathing Water Quality Index, BWQI) Based on Escherichia coli Physiological States for Bathing Waters Monitoring." Journal of Marine Science and Engineering 9.2 (2021): 120.) and related references.

Reviewer 2 Report

An useful and rigorous review in the specific field of recreational waters.

References represent a main component of a review and could be further improved or updated:

* Regarding similar studies and in particular comparisons in Europe within mediterrean countries: Mavridou A, et al. An overview of pool and spa regulations in Mediterranean countries with a focus on the tourist industry. J Water Health. 2014 Sep;12(3):359-71. doi: 10.2166/wh.2014.188. PMID: 25252339.

* Regarding the peculiar issue of natural ponds (bio-pools) in Europe: Giampaoli S, et al. Regulations concerning natural swimming ponds in Europe: considerations on public health issues. J Water Health. 2014 Sep;12(3):564-72. doi: 10.2166/wh.2014.211. PMID: 25252360.

* At lines 35-37: Additional references to be considered may include: (Quilliam et al. The disparity between regulatory measurements of E. coli in public bathing waters and the public expectation of bathing water quality. J Environ Manage. 2019  doi: 10.1016/j.jenvman.2018.11.138; Giampaoli S et al. Health and safety in recreational waters. Bull World Health Organ. doi: 10.2471/BLT.13.126391).  

* at lines 166-174: regarding techniques used for Enteroccocci and E. coli a note on fast methods and/or a molecular approach could be considered:  (e.g. Shrestha et al. Fecal pollution source characterization at non-point source impacted beaches under dry and wet weather conditions. Water Res. 2020 doi: 10.1016/j.watres.2020.116014; Sivaganesan et al. Standardized data quality acceptance criteria for a rapid Escherichia coli qPCR method (Draft Method C) for water quality monitoring at recreational beaches. Water Res. 2019  doi: 10.1016/j.watres.2019.03.011; Staley et al. Microbial Source Tracking Using Quantitative and Digital PCR To Identify Sources of Fecal Contamination in Stormwater, River Water, and Beach Water in a Great Lakes Area of Concern. Appl Environ Microbiol. 2018 doi: 10.1128/AEM.01634-18; Dorevitchet al. Monitoring urban beaches with qPCR vs. culture measures of fecal indicator bacteria: Implications for public notification. Environ Health. 2017  doi: 10.1186/s12940-017-0256-y; Valeriani et al. The molecular enrichment approach for the identification of microbiological indicators in recreational waters. 2014 10.1016/J.MICROC.2013.09.013). 

* Please, verify the all the acronyms and that all are well described in the text.
